# Cerebello-thalamo-cortical hyperconnectivity as a state-independent functional neural signature for psychosis prediction and characterization

Hengyi Cao[1], Oliver Y. Chén[1], Yoonho Chung[1], Jennifer K. Forsyth[2], Sarah C. McEwen[3], Dylan G. Gee[1], Carrie E. Bearden[3], Jean Addington [4], Bradley Goodyear[5], Kristin S. Cadenhead[6], Heline Mirzakhanian[6], Barbara A. Cornblatt[7], Ricardo E. Carrión[7], Daniel H. Mathalon[8], Thomas H. McGlashan[9], Diana O. Perkins[10], Aysenil Belger[10], Larry J. Seidman[11], Heidi Thermenos[11], Ming T. Tsuang[6], Theo G.M. van Erp [12], Elaine F. Walker[13], Stephan Hamann[13], Alan Anticevic[9], Scott W. Woods[9] & Tyrone D. Cannon[1,9]

Understanding the fundamental alterations in brain functioning that lead to psychotic disorders remains a major challenge in clinical neuroscience. In particular, it is unknown whether any state-independent biomarkers can potentially predict the onset of psychosis and distinguish patients from healthy controls, regardless of paradigm. Here, using multi-paradigm fMRI data from the North American Prodrome Longitudinal Study consortium, we show that individuals at clinical high risk for psychosis display an intrinsic "trait-like" abnormality in brain architecture characterized as increased connectivity in the cerebello–thalamo–cortical circuitry, a pattern that is significantly more pronounced among converters compared with non-converters. This alteration is significantly correlated with disorganization symptoms and predictive of time to conversion to psychosis. Moreover, using an independent clinical sample, we demonstrate that this hyperconnectivity pattern is reliably detected and specifically present in patients with schizophrenia. These findings implicate cerebello–thalamo–cortical hyperconnectivity as a robust state-independent neural signature for psychosis prediction and characterization.

[1] Department of Psychology, Yale University, New Haven, CT 06511, USA. [2] Department of Psychology, University of California Los Angeles, Los Angeles, CA 90095, USA. [3] Department of Psychiatry and Biobehavioral Sciences, University of California Los Angeles, Los Angeles, CA 90095, USA. [4] Department of Psychiatry, University of Calgary, Calgary T2N 1N4, Canada. [5] Departments of Radiology, Clinical Neuroscience and Psychiatry, University of Calgary, Calgary T2N 1N4, Canada. [6] Department of Psychiatry, University of California San Diego, San Diego, CA 92093, USA. [7] Department of Psychiatry Research, Zucker Hillside Hospital, Glen Oaks, NY 11004, USA. [8] Department of Psychiatry, University of California San Francisco, San Francisco, CA 94143, USA. [9] Department of Psychiatry, Yale University, New Haven, CT 06510, USA. [10] Department of Psychiatry, University of North Carolina, Chapel Hill, NC 27599, USA. [11] Department of Psychiatry, Beth Israel Deaconess Medical Center, Harvard Medical School, Boston, MA 02115, USA. [12] Department of Psychiatry and Human Behavior, University of California Irvine, Irvine, CA 92697, USA. [13] Department of Psychology, Emory University, Atlanta, GA 30322, USA. Correspondence and requests for materials should be addressed to H.C. (email: hengyi.cao@yale.edu) or to T.D.C. (email: tyrone.cannon@yale.edu)

Understanding the fundamental alterations in brain functioning that underlie schizophrenia (SZ) and pinpointing potential neural markers predictive of psychosis in individuals at risk remain major challenges in clinical neuroscience. The heterogeneity of clinical symptoms and associated cognitive and behavioral deficits in patients has motivated a broad search for neurobiological underpinnings across different functional domains[1]. Accordingly, functional magnetic resonance imaging (fMRI) studies on psychosis have used an extensive repertoire of paradigms, including resting state[2–4], working memory[5,6], episodic memory[7], language processing[8,9], emotional processing[10,11], motor control[12], and response inhibition[13] among others. These studies have provided evidence that dysfunction in several brain regions, including prefrontal cortex[2,3,6–10,13], parietal cortex[5,6], medial temporal cortex[5–8,10], cingulate cortex[6,13], thalamus[2–4,10,11], striatum[3,9,11], and cerebellum[2,4,12] may in some way mark risk for and/or expression of psychosis. Nevertheless, the patterns of altered brain function associated with SZ are not entirely consistent across studies[11,14]. In part, this inconsistency is likely to reflect the fact that each study is designed to probe particular brain systems based on particular task features. The goal of the present study was to determine whether a more consistent functional imaging biomarker of psychosis risk will emerge using a strategy that explicitly assesses patterns of brain functioning across paradigms.

Several lines of evidence make the exploration of such a neural signature plausible. From a clinical perspective, although psychotic disorders present a diversity of phenotypes, similar symptoms such as hallucinations, delusions, and disorganized thought and behavior are the core features that distinguish psychosis from other mental disorders[15]. Thus, an intrinsic neural deficit that is closely related to these features may typify individuals with psychosis and with higher vulnerability to psychosis. From a neurobiological perspective, recent work has suggested that the brain possesses an intrinsic and state-independent functional architecture, and functional networks during different paradigms are shaped primarily by this "standard" architecture and secondarily by paradigm-specific features[16–18]. These findings raise the possibility that the neural alterations associated with psychosis may occur in relation to the "standard" network architecture to a greater degree than the state-dependent networks. Further, despite the heterogeneity in results derived from different paradigms, the brain regions that are associated with psychosis highly overlap in the literature[14,19], suggesting the possibility that shared neural mechanisms underlie various functional states. However, such shared mechanisms may be masked by paradigm-specific effects on brain function and thus may be missed when studying each paradigm separately.

Here, using multi-paradigm fMRI data from two independent cohorts, we investigated whether common functional network abnormalities shared across different paradigms mark the risk for psychotic disorders and predict the onset of psychosis among those in a prodromal state. We also examined whether the same network abnormalities are characteristic of patients with SZ. Using principal component analysis (PCA) combined with connectome-wide network-based statistics (NBS), we sought to delineate network-level changes in the human functional connectome that precede conversion to psychosis in a sample of 182 subjects at clinical high risk (CHR) for psychosis (among whom 19 cases later converted to full psychosis during follow-up) and 120 healthy controls drawn from the second phase of the North American Prodromal Longitudinal Study (NAPLS-2) consortium[20]. The observed network alterations in converters, by virtue of their temporal priority, are likely to reflect a state-independent neural trait that leads to the onset of psychosis[21]. As a test of the robustness of the resulting biomarker, we further

examined the presence and specificity of the finding in a second cohort including 50 patients with SZ, 49 patients with bipolar disorder (BD), 40 patients with attention deficit hyperactivity disorder (ADHD), and 123 healthy controls drawn from the Consortium for Neuropsychiatric Phenomics (CNP) study[22]. We hypothesized that (1) there would be a common connectomic signature of psychosis risk across different brain functional states and in particular those who later converted to psychosis; and (2) this state-independent connectomic profile related to psychosis risk would be specifically present in patients with SZ and not observed among patients with other forms of psychiatric illness.

## Results

**PCA for the NAPLS-2 data.** All subjects in the NAPLS-2 sample (302 in total, including 19 converters, 163 non-converters, 120 controls) completed a battery of five fMRI paradigms at the point of recruitment: an eyes-open resting-state paradigm, a verbal working memory task, an episodic memory encoding task, an episodic memory retrieval task, and an emotional face matching task. We used the expanded Power brain atlas with 270 regions[23–25] to construct functional brain networks for each individual during each paradigm, thereby generating a total of $302 \times 5$ whole-brain connectivity matrices, each representing the pairwise connectivity between the 270 nodes for a given subject and paradigm. To ascertain the existence of a common brain functional architecture independent of paradigm[16,17], we first performed a PCA analysis on the constructed connectivity matrices, aiming to extract the shared connectivity patterns that can explain the majority of variance across all paradigms for each individual (Fig. 1). We found that for all three studied groups, the first principal component (PC) scores explained ~70% of the total variance in the connectivity matrices across all five paradigms (Supplementary Fig 1A). There were no significant differences in percent of variance explained between groups ($P = 0.16$, one-way ANOVA). In addition, when examining each paradigm separately, we found that the resting state, working memory, episodic memory encoding, and emotional face matching paradigms showed similar factor loadings on the first PCs, while the episodic memory retrieval paradigm had a slightly lower loading, suggesting a relatively smaller contribution of the memory retrieval paradigm to the first PCs compared with other paradigms. However, no significant group differences were found in factor loadings for each of the paradigms ($P > 0.44$, one-way ANOVA, Supplementary Fig 1B), suggesting that all three groups had paradigm-wise similar contributions to the first PC.

**NBS for the first PCs in the NAPLS-2 data.** After confirming that the first PC matrices can explain the majority of variance across paradigms and thus can serve as a "state-independent" trait matrix for each individual, we next considered whether there were any connectivity changes within these PC matrices between groups. Importantly, although not a direct measure of "functional connectivity" as defined traditionally using correlation-based methods, the values in a PC matrix do reflect the strength of functional connectivity shared across all paradigms for a given individual. Here we termed these values as measures of "Cross-paradigm connectivity", in order to differentiate them from "functional connectivity" in a more typical context. Here, NBS was employed to examine this question following established procedures used in prior studies[11,24,26]. Notably, in addition to variance from neural signals, the first PC matrices derived from the PCA analysis could also capture signals associated with subjects' demographic features, head motion, and/or medication status, since the variations related to these variables are also consistently present across paradigms. To mitigate these

confounding influences, we included age, sex, IQ, site, mean frame-wise displacement (FD) across all paradigms, and antipsychotic dosage as nuisance regressors in the NBS analysis. After controlling for these variables, we observed a highly significant group effect on a connected network including a total of 84 edges linking pairs of 62 nodes covering multiple brain regions in the cerebellum, thalamus, and cerebral cortex ($P_{FWE} = 0.005$ from 10,000 permutations, Fig. 2a). In particular, the regions in the identified network belonged to seven functional systems as previously defined[23]: subcortical-cerebellar (e.g., thalamus, putamen, cerebellum), sensorimotor (e.g., pre- and postcentral gyri, supplementary motor area), visual (e.g., middle and inferior occipital gyri, inferior temporal gyrus, lingual gyrus, fusiform gyrus), auditory (e.g., rolandic operculum), default-mode (e.g., medial prefrontal gyrus, angular gyrus, precuneus, middle temporal gyrus), frontoparietal (e.g., superior and middle frontal gyri), and attentional (e.g., superior and middle temporal gyri). The PC scores representing the cross-paradigm connectivity between these regions were significantly higher in subjects at CHR compared with controls, an effect that was significantly more pronounced in those who later converted to psychosis than non-converters (Fig. 2b), suggesting a paradigm-independent connectivity alteration that precedes onset of psychosis.

To better interpret the NBS findings, we further investigated two questions. First, since the signs of values in the PC matrices have been rescaled and might not be the same as in the original correlation matrices, whether the higher cross-paradigm connectivity observed in converters indeed reflected hyperconnectivity was unclear. Second, it was unknown whether the detected effect was driven by any particular paradigms. To answer these questions, the entire identified network was extracted from the original connectivity matrices for each paradigm and averaged across all edges in this network. We found a significant group effect for all five paradigms on the mean functional connectivity of this network ($P_{FWE} < 0.04$, one-way ANCOVA, Fig. 2c). Similarly, the converters showed the highest connectivity, followed by the non-converters, while the control subjects had the lowest connectivity. In addition, the functional connectivity measures in all three groups were positive. These findings suggest a cerebello–thalamo–cortical hyperconnectivity in converters that is not driven by particular paradigms but rather, present in all paradigms used in the study.

**Association with psychosis severity**. To examine potential associations between the identified network alteration and the severity of psychosis symptoms, we performed Spearman rank-order correlations between the mean network cross-paradigm connectivity and the positive and disorganization scores acquired from the Scale of Prodromal Symptoms (SOPS[27]). Notably, positive and disorganization symptoms are diagnostically more specific to psychosis than negative and general symptoms. We observed a significant association of the network measure with the disorganization symptoms in subjects at CHR ($R = 0.17$, $P = 0.02$, Fig. 2d) but not in healthy controls ($P = 0.41$). The correlation between the network measure and positive symptoms did not reach significance in either group ($P > 0.12$). These findings suggest that the observed hyperconnectivity may be related to bizarre thought and behavior in individuals with prodromal symptoms.

**Association with psychosis conversion speed**. We then investigated whether the observed network alteration that preceded the onset of psychosis would predict the time to conversion in CHR converters. To that end, Spearman rank-order correlation was performed between the mean network cross-paradigm

connectivity and the number of months to conversion after the baseline scan. We observed a significant correlation between these two variables ($R = -0.48$, $P = 0.04$, Fig. 2e), suggesting that higher connectivity in the cerebello–thalamo–cortical network predicts shorter conversion time.

**Association with structural measures**. Since the observed connectivity changes in the cerebello–thalamo–cortical circuit are robust across different paradigms, a question naturally arises as whether these changes relate to structural differences in identified nodes in this circuit, in which case the connectivity metrics may be redundant with anatomical measures in indexing risk for psychosis. To address this question, we extracted gray matter volumes of all identified cortical, subcortical, and cerebellar regions from subjects' processed T1-weighted imaging data and correlated these measures with the mean PC scores of the identified network using Pearson Correlation. Our analysis revealed no significant associations between the functional connectivity measures and the structural gray matter volumes after multiple correction ($P_{FWE} > 1$). The only trend-level effects were shown in bilateral thalamus ($R = -0.12$, $P_{uncorrected} = 0.04$), suggesting that the observed cerebello–thalamo–cortical hyperconnectivity conveys unique information on risk for psychosis that is not fully explained by anatomical changes associated with psychosis, and/or may occur at the time point before the most pronounced structural changes appear.

**Verification of results in a matched subsample**. To confirm that the detected network change was not explained by demographic and/or clinical variables on which there were also significant group differences (Supplementary Table 1), we performed a supplementary analysis using a small subsample of subjects in the NAPLS-2 cohort that were unmedicated and well matched in terms of demographics across outcome groups (see Supplementary Table 2). The subsample included a total of 11 converters, 40 non-converters, and 40 healthy controls drawn from the larger sample reported above. Here, same as in the larger sample, we observed significant group differences in the cross-paradigm connectivity of the identified network ($P < 0.001$, one-way ANCOVA, Supplementary Fig 3A). Again, the highest values were shown in converters, followed by non-converters and controls. These data further verify that the detected hyperconnectivity pattern in converters is not driven by group differences in demographics and medication.

**Comparison between subjects with 24-month clinical follow-up**. Since clinical follow-up time in the NAPLS-2 sample varied between individuals, and those with relatively short duration of follow-ups were more likely to include persons who actually ended up converting, we compared the mean cross-paradigm connectivity of the identified network between CHR converters and CHR non-converters that had been followed-up for at least 24 months in a supplementary analysis (19 converters and 103 non-converters). Similar to the result in the whole sample, this supplementary analysis showed a significant group difference between converters and non-converters ($P = 0.004$, one-way ANCOVA). Moreover, larger effect size (Cohen's $d = 0.76$) was observed in this subsample compared with that in the whole sample (Cohen's $d = 0.68$), suggesting that the observed hyperconnectivity in the NAPLS-2 sample may actually be underestimated.

**Specificity of the observed network**. Since the identified network included a total of 84 edges, the relatively large size of this network raises the question as whether such change was

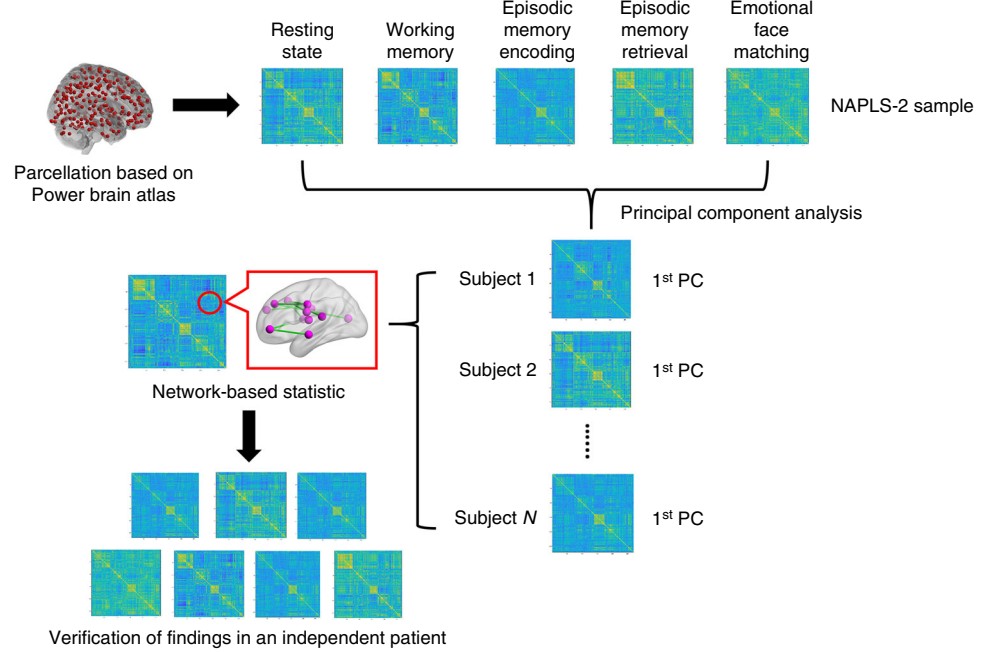

**Fig. 1** Flowchart of the processing pipeline used in this study

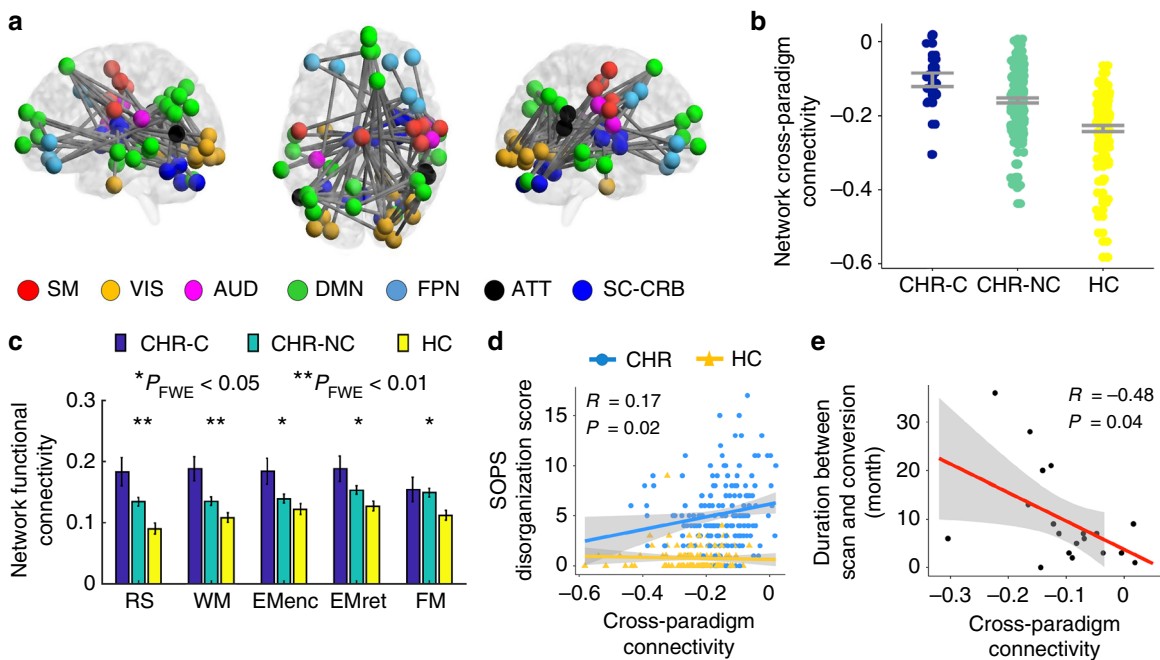

**Fig. 2** Network alteration observed in the NAPLS-2 data. **a** The identified network with higher connectivity in converters and non-converters compared with controls from the NBS analysis. The nodes in the network mapped to seven functional systems (SM sensorimotor, VIS visual, AUD auditory, DMN default-mode, FPN frontoparietal, ATT attentional, SC-CRB subcortical-cerebellar). **b** Significant linear relationship was shown for the mean cross-paradigm connectivity of the identified network between three groups, with the converter group having the highest value and the control group having the lowest. Note that the cross-paradigm connectivity values were defined at the PCA space, which was rescaled to be mean centered at zero. CHR-C converters, CHR-NC non-converters, HC healthy controls. **c** The functional connectivity strength of the identified network in the original connectivity matrices for three groups. Significant effects were shown for all five paradigms (RS resting state, WM working memory, EMenc episodic memory encoding, EMret episodic memory retrieval, FM emotional face matching). **d** The mean cross-paradigm connectivity of the network was significantly correlated with the SOPS disorganization scores in subjects at clinical high risk but not in healthy controls. **e** The mean cross-paradigm connectivity of the network significantly predicted time to conversion to psychosis among converters. Error bars indicate standard errors

edge-specific or rather generic across the whole brain. Here, we performed an additional permutation test to examine the specificity of the identified network. Specifically, during each permutation, we randomly selected 84 edges from the PC matrices and compared the group differences on the means of these selected edges. The whole procedure was iterated 10,000 times. We found that none of the $P$ values derived from the 10,000 permutations reached statistical significance after Bonferroni correction (Supplementary Fig. 4). In stark contrast, the observed network was highly significant even after Bonferroni correction for the 10,000 permutations. This supplementary analysis supports the specificity of the identified network in psychosis prediction, demonstrating that it is not driven by effects at the global level.

**NBS analysis on the resting-state data.** To assess whether the observed network hyperconnectivity was simply a reflective of resting-state abnormality (in which case the PCA analysis would be redundant), we performed an additional NBS analysis solely on the resting-state data. This analysis revealed no significant differences between the outcome groups, suggesting that the observed network change is detectable only when collapsing across multiple paradigms rather than during rest .

**Association with head motion parameters.** To further ensure that the detected network abnormality was not driven by head motion differences between groups, we performed an additional analysis to test the potential association between the observed network metrics and frame-wise displacement values across all individuals in the NAPLS-2 sample using Spearman rank-order correlation. This analysis revealed no significant correlation between the two variables ($R = 0.08$, $P = 0.17$), which supports the argument that the detected network abnormality is unlikely to be driven by head motion differences between groups.

**Presence of network hyperconnectivity in the CNP data.** To confirm that the detected network hyperconnectivity is a "trait" abnormality for psychosis, we further investigated the presence of such alteration in an independent sample with multi-paradigm fMRI data acquired from three clinical populations (SZ, BD, and ADHD) and healthy controls (Supplementary Table 3). The subjects in the CNP sample completed some or all of the seven paradigms employed by the cohort: an eyes-open resting-state paradigm, a "balloon-analog" risk taking task, a spatial working memory task, an episodic memory encoding task, an episodic memory retrieval task, a "Go–No Go" stop signal task, and a

"color-shape" task-switching task. Following the same procedures described above, we computed the first PC scores for the correlation matrices across all paradigms and extracted the values from the same network for each individual (Supplementary Fig 2). As expected, we observed a significant group effect on the network cross-paradigm connectivity after controlling for age, sex, IQ, mean FD, and antipsychotic dosage ($P = 0.025$, one-way ANCOVA, Fig. 3a). Specifically, this effect was driven by the differences between the SZ group and the HC group ($P_{Bonferroni} = 0.024$, post-hoc $t$-test) but not between the other groups ($P_{Bonferroni} > 0.26$, post-hoc $t$-test). Moreover, there tended to be a gradient elevation of the degree of hyperconnectivity in the identified network with the increase of prevalence of psychotic symptoms in the populations (such that SZ > BD > ADHD > HC). These findings suggest a psychosis-specific functional neural signature in patients, in particular those with SZ.

To further verify the association between the network hyperconnectivity and the disorganization symptoms as identified in the NAPLS-2 sample, Spearman rank-order correlations were performed for the network cross-paradigm connectivity measures on each of the four subscales (hallucinations, delusions, bizarre behavior, thought disorder) of the Scale for the Assessment of Positive Symptoms (SAPS[28]) in patients with SZ. Consistent with the finding in the NAPLS-2 sample, the result revealed a significant correlation between the network measure and the thought disorder subscale scores ($R = 0.30$, $P = 0.035$, Fig. 3b). The correlations with other subscales did not reach statistical significance ($P > 0.30$), suggesting that the observed network alteration may be specifically related to disorganized thought and speech in patients.

Similar to the procedures used in the NAPLS-2 sample, we also confirmed the findings in a demographically matched subsample of the CNP cohort including 27 patients with SZ, 27 patients with BD, 27 patients with ADHD, and 27 HCs (Supplementary Table 4). The same group effect was again identified ($P = 0.016$, one-way ANCOVA, Supplementary Fig 3B), which was again driven by the differences between the SZ group and the HC group ($P_{Bonferroni} = 0.043$, post-hoc $t$-test) but not between the other groups ($P_{Bonferroni} > 0.06$, post-hoc $t$-test). These findings suggest that the detected connectivity differences in the larger sample are unlikely to be a result of unmatched demographics between groups. Encouraged by these results, we further performed a receiver operating characteristic (ROC) curve analysis to test the ability of using the hyperconnectivity pattern discovered in the NAPLS-2 data to distinguish patients with SZ from the controls in the overall CNP sample. Our analysis revealed an area under

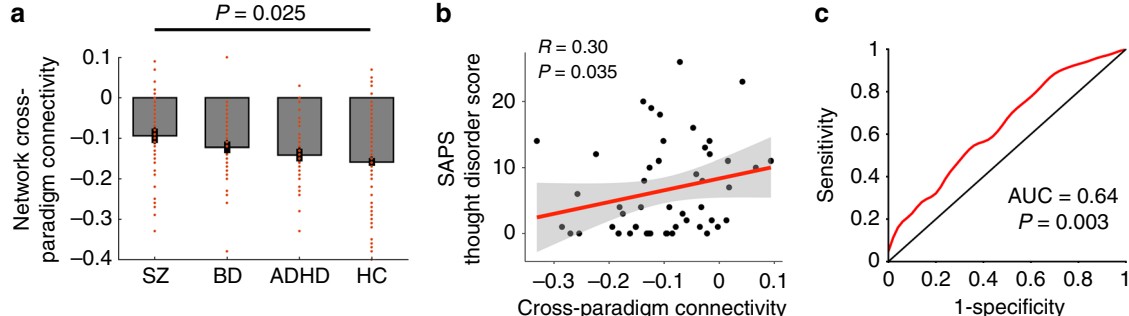

**Fig. 3** The presence of the observed network alteration in the CNP data. **a** Significant group differences were shown for the mean cross-paradigm connectivity of the identified network, which was driven by the differences between schizophrenia and controls. SZ schizophrenia, BD bipolar disorder, ADHD attention deficit hyperactivity disorder, HC healthy control. **b** The network alteration was significantly correlated with scores of SAPS thought disorder subscale in patients with schizophrenia. **c** Receiver operating characteristic curve for distinguishing patients with schizophrenia from healthy controls. The area under curve was significantly higher than that can be achieved by chance, per permutation testing. Error bars indicate standard errors

curve (AUC) of 0.64 ($P = 0.003$ from 10,000 permutations, Fig. 3c), further supporting a trait hyperconnectivity alteration that can potentially be used for psychosis prediction and characterization.

## Discussion

What are the fundamental brain functional alterations that lead to psychosis? Although prior work has converged on the hypothesis of multiple target regions in the cerebral cortex, subcortex, and cerebellum[2,14,19], direct evidence that binds these discrete observations into a unified framework is still lacking. In this study, we combined PCA and NBS approaches to analyze data from two independent cohorts to show that cerebello–thalamo–cortical hyperconnectivity is a "trait" alteration that can be robustly detected across different fMRI paradigms in subjects with psychosis and with high vulnerability to psychosis. These findings suggest a state-independent functional neural signature that precedes and potentially predicts psychotic disorders.

The results of our study extend existing knowledge in clinical neuroscience and offer some useful implications for psychosis research going forward. First, our data provide the first empirical evidence that psychosis is associated with an intrinsic "trait-like" abnormality in functional brain architecture, which occurs before the onset of full illness. This "trait-like" abnormality involves dysconnectivity between multiple cerebral cortical regions, thalamus, and cerebellum. The affected cortical regions correspond well to those that have been frequently reported as associated with SZ in the literature, such as the medial prefrontal cortex[3,7,29], dorsolateral prefrontal cortex[2,3,6–10,29], orbitofrontal cortex[29,30], medial temporal cortex[7,8,10,30], lateral temporal cortex[3,5], parietal cortex[2,5,6], and occipital cortex[10,31]. The associations of these regions with psychosis have been found both anatomically and functionally, including measurements such as cortical thickness[29], cortical volume[19], brain activation and connectivity during tasks[5–8,10] and rest[2–4,12], and low-frequency brain oscillations[31], suggesting that the changes in these regions are robust and may be consistently present across multiple imaging modalities. Our study extends these previous results by showing the existence of a state-independent network anomaly, suggesting that circumscribed regional alterations reported previously may partly belong to a broader and intrinsic neural network change in psychosis. See Supplementary Discussion for further discussion of the relation of these findings to previous results.

From a systems neuroscience perspective, the nodes in the detected network belong to seven functional systems that are strongly associated with psychotic disorders. Specifically, the frontoparietal and attentional networks are critical cognitive control systems in humans[32,33] and have been implicated in the cognitive deficits in patients with psychosis, particularly SZ[34,35]. The default-mode network is a brain system that is activated during rest but deactivated during attention-demanding tasks[36]. The failure to deactivate this network during active tasks and the exaggeration of connectivity in this network during resting state have been repeatedly reported in the psychosis state[30,37,38], which may relate to excessive internally focused thoughts and self-reference during rest and lack of sufficient suppression of these thoughts during task[39]. The dysfunction in primary functional systems such as the visual and auditory networks may contribute to hallucination symptoms in psychotic disorders[40–42], and the altered connectivity between thalamus and the sensorimotor network may reflect a subcortical gating deficit which leads to abnormal subcortical sensory inputs to the cortex[2,43–46]. Together, the conjoint involvement of these systems in the identified network change suggests aberrant information integration or communication between multiple primary and higher-order systems as a core feature of psychosis. The effects of each system may summate and/or interact with each other in creating risk for overt illness.

Our findings appear to be broadly consistent with the "cognitive dysmetria" theory of SZ[47]. Initially proposed by Nancy Andreasen and colleagues[47–49], this theory posits that patients with SZ are characterized by changes in a key neural circuit, namely, the cerebello–thalamo–cortical circuitry. Dysfunction in this circuitry leads to an impairment in the synchrony or coordination of mental processes. This impaired mental coordination is considered as the fundamental deficit in SZ that further accounts for various clinical symptoms. This theory is supported by prior neuroimaging studies. For example, altered cerebellar and prefrontal activations have been reported during a wide range of cognitive tasks[48,50–52]. In addition, abnormal thalamic–cortical and cerebellar–cortical functional connectivity have been consistently observed in populations with psychosis, particularly SZ[43–46,53,54]. These alterations can further be found before the onset of psychosis in subjects at high risk[4,12,55], suggesting a trait anomaly that possibly relates to vulnerability to psychotic disorders. Apart from these functional findings, structural changes in thalamus, cerebellum, and prefrontal cortex are also evident in SZ[29,56] and have been summarized in several meta- or mega-analysis studies[57–59]. Highly consistent with these prior findings, our research further demonstrates that the dysfunction of the cerebello–thalamo–cortical circuitry may be a fundamental state-independent neural trait that predicts and characterizes psychosis, thereby highlighting the role of this circuitry in the neuropathology and psychopathology of SZ. These findings overall may offer a unified framework to explain the complex and heterogeneous behavioral and cognitive phenotypes in psychotic disorders.

The hyperconnectivity in the identified network is also broadly consistent with the "N-methyl-D-aspartate receptor (NMDAR) hypofunction" hypothesis of SZ[60–62]. Reduced global NMDAR expression and activity have long been observed in patients with SZ[60,61,63], possibly due to underlying genetic and environmental risk factors[64,65]. These NMDAR deficits would downward regulate the function of cortical parvalbumin-containing γ-aminobutyric acid (GABA) interneurons, leading to reduced recurrent inhibition of the pyramidal glutamatergic neurons, which in turn, cause exaggerated cortical gamma oscillations and functional connectivity in patients[60,66,67]. As a result, the hyperconnectivity in the cerebello–thalamo–cortical circuitry in our study may reflect a downstream phenomenon of NMDAR functional deficits, which occurs prior to the onset of psychosis and maintains during the full psychosis state. Another interpretation, in line with the "cognitive dysmetria" theory, relates to the need for increased cognitive effort in SZ patients, as well as in subjects at CHR. A large body of work has pointed to the role of cerebellum as a general "error detection and correction" center in the brain, which receives, integrates, and computes error information regarding both movement and thought from the cerebral cortex and provides adaptive feedback via the cerebello–thalamo–cortical circuitry[68–70]. In parallel with this notion, the hyperconnectivity in this circuitry may reflect a compensatory effect induced by excessive error input from the upstream cerebral cortex. In other words, individuals with psychosis and at high risk may require more effort in error processing in order to coordinate their behaviors and thoughts.

Our study has some limitations to note. First, the results reported here were based on the PCA analysis of the original connectivity matrices across multiple paradigms, which essentially extracted the individual-specific connectivity characteristics in brain organization. These individual-specific connectivity

characteristics, although largely attributed to biological and (patho)physiological differences, may also be affected by other factors such as demographics, head motion, and medication. Although we sought to control these factors by directly modeling these variables in our analyses and by verifying results in sub-samples matched on these features, we cannot fully exclude the possibility that our findings may to certain degree be influenced by other factors that we have not measured. Second, despite the fact that our study was performed on one of the largest psychosis CHR fMRI sample to date, the number of converters in our NAPLS-2 sample was relatively small. Therefore, future replications are encouraged in larger cohorts with more data available for converters. Third, all patients recruited in the CNP cohort were chronic patients. As a result, the detected connectivity differences between groups may also partly reflect differences in disease chronicity, treatment, comorbidity, among others. Due to the lack of sufficient clinical data in the CNP sample, these factors cannot be isolated and thus may to certain degree contribute to the current findings.

To sum up, using multi-paradigm fMRI data from two independent cohorts, our study provides the first evidence for cerebello–thalamo–cortical hyperconnectivity as a state-independent neural trait for psychosis prediction and characterization. The results revealed in the study converge with established theories of SZ focusing on coordination and timing of cognitive and motor systems as well as excitation–inhibition balance, and may potentially help to advance a more unified framework for understanding of the neuropathology of psychosis. Future research is encouraged to replicate these findings and to investigate the nuanced role of this circuitry in relation to cognitive, symptomatic, and other features of psychotic illness.

## Methods

**Subjects**. The NAPLS-2 sample consisted of 182 subjects at CHR for psychosis (including 19 subjects who converted to psychosis during follow-up (CHR-C) and 163 subjects who did not convert (CHR-NC)) and 120 healthy controls (HC) as part of the NAPLS-2 project[20] recruited from eight study sites across the United States and Canada. All included subjects completed a battery of fMRI scans with five different paradigms (resting state, working memory, episodic memory encoding, episodic memory retrieval, emotional face processing) at the initial recruitment point. The participants provided written informed consent for the study. The protocol and consent forms were approved by the institutional review boards at each site. The CNP sample was drawn from a publicly available dataset (UCLA CNP study[22], https://openfmri.org/dataset/ds000030/). The final sample used in this study included 50 patients with SZ, 49 patients with BD, 40 patients with ADHD, and 123 HCs. The included participants underwent some or all of the seven paradigms employed in the cohort (resting state, risk taking, working memory, episodic memory encoding, episodic memory retrieval, stop signal, task switching). Subjects provided written informed consent following procedures approved by the Institutional Review Boards at UCLA and the Los Angeles County Department of Mental Health. Details on both samples are provided in Supplementary Methods.

**Common data processing for both samples**. Both samples followed the same preprocessing pipelines using the Statistical Parametric Mapping software (SPM12, [http://www.fil.ion.ucl.ac.uk/spm/software/spm12/]). After preprocessing, we extracted the mean time series from each of the 270 nodes in the extended Power atlas as previously defined[23,24]. The extracted time series were further corrected for mean effects of task-evoked coactivations, white matter (WM) and cerebrospinal (CSF) signals, 24 head motion parameters (6 translation and rotation parameters, their first derivatives, and the square of these 12 parameters), and frame-wise displacement (FD), and then temporally filtered (rest data: band-pass 0.008-0.1 Hz; task data: high-pass 0.008 Hz). Pairwise correlations were performed between the noise-corrected, filtered time series of each of the 270 nodes, yielding a $270 \times 270$ connectivity matrix for each subject during each paradigm. To generate a subject-specific "trait" matrix that can explain the majority of variance across paradigms, the first PC scores were extracted from the connectivity matrices across all paradigms for each individual. Details on data processing can be found in Supplementary Methods.

**Network discovery in the NAPLS-2 sample**. We used NBS (NBS, [https://sites.google.com/site/bctnet/comparison/nbs]) to probe connectivity differences in the extracted first PC matrices between the three outcome groups in the NAPLS-2 sample following previous publications[11,24,26]. Age, sex, IQ, site, and mean FD across all paradigms and antipsychotic dosage were included in the model as nuisance regressors to strictly control for potential confounding effects related to subjects' demographic, head motion, and/or medication status. The significance of results was determined by 10,000 permutation testing. For further verification purposes, the mean values of the identified network in the PC matrices were extracted for each individual and tested in an unmedicated, matched subsample using an analysis of covariance (ANCOVA) model including the same nuisance variables described above as covariates. To test potential associations with structural data, subjects' high-resolution T1-weighted images were processed using the standard pipeline in the FreeSurfer software ([https://surfer.nmr.mgh.harvard.edu/]). The gray matter volumes of the regions in the identified network were subsequently extracted and Pearson correlations were performed between these gray matter volumes and the mean values of the network in the PC matrices. Statistical significance was determined at $P < 0.05$ after family-wise error (FWE) correction. See Supplementary Methods for details on these analyses.

**Network verification in the CNP sample**. Following the same procedure as described above, the mean values of the identified network in the PC matrices were extracted for each individual in the CNP sample. Group effects on the identified network were examined using an ANCOVA model with age, sex, IQ, and mean FD across all paradigms and antipsychotic dosage as covariates. The same model was also used for the subsample analysis. Details are provided in Supplementary Methods.

**Code availability**. Data were processed using publicly available software (SPM12, [http://www.fil.ion.ucl.ac.uk/spm/software/spm12/]; NBS toolbox, [https://sites.google.com/site/bctnet/comparison/nbs]; and Freesurfer, [https://surfer.nmr.mgh.harvard.edu/]). The PCA analysis was performed using Matlab inbuilt function *pca*.

## Data availability

The CNP data are publicly available at https://www.legacy.openfmri.org/dataset/ds000030/. The NAPLS-2 data are available from corresponding authors upon request.

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

## Acknowledgements

This work is supported by the NARSAD Young Investigator Grant (No. 27068) to Dr. Cao, by gifts from the Staglin Music Festival for Mental Health and International

Mental Health Research Organization to Dr. Cannon, and by National Institute of Health (NIH) grants U01 MH081902 to Dr. Cannon, P50 MH066286 and the Miller Family Endowed Term Chair to Dr. Bearden, U01 MH081857 to Dr. Cornblatt, U01 MH82022 to Dr. Woods, U01 MH066134 to Dr. Addington, U01 MH081944 to Dr. Cadenhead, R01 U01 MH066069 to Dr. Perkins, R01 MH076989 to Dr. Mathalon, U01 MH081928 to Dr. Seidman, and U01 MH081988 to Dr. Walker.

## Author contributions

H.C. and T.D.C. conceptualized the study; C.E.B., J.A., K.S.C., B.A.C., D.H.M., T.H.M., D.O.P., L.J.S., M.T.T., E.F.W., S.W.W., and T.D.C. designed and organized the whole NAPLS consortium; S.C.M., C.E.B., J.A., K.S.C., B.A.C., D.H.M., T.H.M., D.O.P., L.J.S., M.T.T., E.F.W., S.W.W., T.D.C., B.G., H.M., R.E.C., A.B., H.T., T.G.M.E., and S.H. collected the data; H.C., O.Y.C., Y.C., J.K.F., S.C.M., D.G.G., and A.A. analyzed the data; and H.C. and T.D.C. drafted the paper with comments from all authors.

## Additional information

**Competing interests:** T.D.C. has served as a consultant for Boehringer–Ingelheim Pharmaceuticals and Lundbeck A/S. The remaining authors declare no competing interests.

