## [Peer Review File · Nature Communications]

Reviewers' comments:

Reviewer #1 (Remarks to the Author):

The paper by Cao et al. describes a functional connectivity study in a sample of 182 clinical high-risk subjects (of whom 19 later converted to psychosis) and 120 healthy controls. Using a principal component analysis, the researchers condensed information from several fMRI paradigms including resting-state and task-fMRI into one functional connectivity matrix that likely reflects underlying functional connectivity patterns that exist regardless of a given functional state. This connectivity matrix was then analyzed using network-based statistic (NBS) to identify clusters or subnetworks of functional connections showing group-differences. The study has some interesting implications because it shows that functional connectivity abnormalities can be identified prior to psychosis onset in at-risk individuals and are independent of fMRI paradigm, and I am impressed that the authors performed a replication of their findings in an independent sample of schizophrenia patients. I also have some concerns, however, mainly regarding the novelty and specificity of the findings.

First, while it is an interesting approach to derive a functional connectivity matrix from different fMRI paradigms, it appears that the findings would have been similar if the authors had simply analyzed the rs-FC matrix. When the NBS-network was retested in fMRI paradigms separately, findings were similar across paradigms, with the group-effect appearing strongest in the rs-network (fig 2C). In this sense, the study seems similar to many previous functional connectivity studies in schizophrenia and high-risk samples, including studies applying network-based statistic analysis (e.g., Fornito et al., 2011; Brent et al., 2017; Ray et al., 2017; Zhu et al., 2016), and I wonder if the findings are sufficiently novel to warrant publication in Nature Communications.

Second, I am concerned about the specificity of the identified subnetwork. In the NBS-analysis, the authors use a rather strict $p < 0.0005$ threshold to identify suprathreshold links and still find an extended network of 84 edges among 62 nodes encompassing a large part of the brain. Is there possibly a global connectivity effect driving this result, rather than a specific abnormality of the links in the reported subnetwork? If potential global connectivity differences are mitigated, is there still a significant group-difference in functional connectivity of the NBS-network? Also, if a permutation analysis is performed in which clusters of 84 edges are repeatedly selected at random, how often does this result in significant group-effects?

Third, the authors report increased functional connectivity between cerebellum, thalamus, and cerebral cortex in CHR subjects and SZ patients. Reviews on functional connectivity in schizophrenia suggest that FC is more commonly found to be reduced, or to involve a combination of increased and decreased FC, in patients relative to controls (e.g., Petterson-Yeo et al., 2011; Fornito et al., 2012), a notable example being a previous publication by the same group reporting increased connectivity between thalamus and sensorimotor cortex, but decreased connectivity among thalamus, prefrontal cortex and cerebellum in the NAPLS 2 sample (Anticevic et al., 2015). The authors do not address the issue of increased versus decreased functional connectivity in schizophrenia/high-risk in general and in relation to cerebello-thalamo-cortical connectivity specifically. Can the authors expand on this issue and comment on possible reasons that they find increased connectivity when decreased connectivity (or a mixture of increased and decreased connectivity) is more often reported? Are there methodological factors that may explain the direction of the effects?

Reviewer #2 (Remarks to the Author):

This article deal with an analysis of functional connectivity in a wide dataset of psychiatric disorders, including converter patients . The results are then replicated in a different dataset, also

including psychiatric disorders. Overall the article is clearly written, and the goals and results well presented. Being more familiar with the methodological part than the clinical part, my questions and remarks will mostly focus on the methodological aspect of the manuscript .

- First of all, I do not I fully agree with the denomination "cross-paradigm connectivity" used throughout the manuscript. It appear that such a denomination would involve some 'paradigm-specific' connectivity, whereas it appears the way the analysis are done results in exactly the opposite, i.e. get rid of everything linked to the paradigms by regressing out the task regressors. Please justify this denomination, or use another term.

- Apart from the denomination, I am also not sure of the meaning from a methodological point of view of correlating the residuals of a task-based experiment . It would appear to me that all the task-based paradigm where task regressors have been regressed would corresponds to some 'resting-state' effect, i.e. where the focus is based on the baseline rather than the task part. However, it will depend on the design of each specific experiment to assess whether this is really what is measured here: if for example the task consists of a very fast repetition of stimuli over the entire run, then the baseline will be very poorly modeled, and thus the residuals after task regression will consist mostly of noise. Please justify the quality of the baseline evaluation in order to use the method on the chosen paradigms.

-Finally, it is necessary (as always in functional connectivity) to be extra cautious about the head movement. Here it appears that extra care has been taken in the computation of the connectivity matrices; however it appears that there is a significant difference in the head motion between the groups in the first dataset (table S1, last line). Is the head motion included when testing for differences in overall connectivity between the groups (for example in Figure 2B, or even 2C)? Head motion may indeed lead to higher values in functional connectivity. Please make sure this effect are taken into account in all analyses.

- And a last remark: I am not fully satisfied with the term 'Cerebello-thalamo-cortical': although I understand that it may come from theories in psychiatry, the results shown here do not involve only connection between one of this subpart of the brain: many connections found here between areas within the cortex. Also it make sense to when talking about thalamo-cortical loops (i.e. involving connections between the cortex and the thalamus and the reverse), or even cerebello-thalamo-cortical loops, 'Cerebello-thalamo-cortical' may as well refer to 'whole brain' (apart from some sub-cortical parts of it). Please change the denomination or justify it.

Minor notes:

- The reference 17 (DSM-5) appears very strangely formatted.
- In the results part, just before the discussion: "The same group effect was again identified ($P = 0.016$, Fig S3B), which was again driven by the differences between the SZ group and the HC group ($P_{\text{Bonferroni}} = 0.043$) but not between the other groups ($P_{\text{Bonferroni}} > 0.06$).". Do the last stats involve separately different pairs of groups? Why is there a single value in this case?

Response to reviewers

To the reviewers,

We would like to express our sincere gratitude to both of you for your constructive and valuable suggestions, which have helped to improve this paper significantly. We have revised the paper per your suggestions, as detailed below.

With our best regards,

Hengyi Cao & Tyrone Cannon
On behalf of all coauthors

Reviewer #1 (Remarks to the Author):

The paper by Cao et al. describes a functional connectivity study in a sample of 182 clinical high-risk subjects (of whom 19 later converted to psychosis) and 120 healthy controls. Using a principal component analysis, the researchers condensed information from several fMRI paradigms including resting-state and task-fMRI into one functional connectivity matrix that likely reflects underlying functional connectivity patterns that exist regardless of a given functional state. This connectivity matrix was then analyzed using network-based statistic (NBS) to identify clusters or subnetworks of functional connections showing group-differences. The study has some interesting implications because it shows that functional connectivity abnormalities can be identified prior to psychosis onset in at-risk individuals and are independent of fMRI paradigm, and I am impressed that the authors performed a replication of their findings in an independent sample of schizophrenia patients. I also have some concerns, however, mainly regarding the novelty and specificity of the findings.

We thank the reviewer for your interest in our work! We have addressed your concerns in the revised paper, detailed below.

First, while it is an interesting approach to derive a functional connectivity matrix from different fMRI paradigms, it appears that the findings would have been similar if the authors had simply analyzed the rs-FC matrix. When the NBS-network was retested in fMRI paradigms separately, findings were similar across paradigms, with the group-effect appearing strongest in the rs-network (fig 2C). In this sense, the study seems similar to many previous functional connectivity studies in schizophrenia and high-risk samples, including studies applying network-based statistic analysis (e.g., Fornito et al., 2011; Brent et al., 2017; Ray et al., 2017; Zhu et

al., 2016), and I wonder if the findings are sufficiently novel to warrant publication in Nature Communications.

We agree this is an important issue. Indeed, judging from Fig 2, although the effect was shown for all examined paradigms, the resting-state paradigm seemed to have the strongest effect. To assess whether the observed network hyperconnectivity was simply a reflective of resting-state abnormality (in which case the PCA analysis would be redundant), we performed an additional NBS analysis on the resting-state data. This analysis revealed no significant differences between the outcome groups, suggesting that the observed network hyperconnectivity is detectable only when collapsing across multiple paradigms rather than during resting state. This finding argues against the possibility that our results are simply a repeat of previous work and support the novelty of our discoveries as elucidated by the novel application of the cross-paradigm PCA methodology. We have now included this additional analysis in the Supplementary Materials.

[Supplementary Results, Page 7, NBS analysis on the resting-state data]: “To assess whether the observed network hyperconnectivity was simply a reflective of resting-state abnormality (in which case the PCA analysis would be redundant), we performed an additional NBS analysis on the resting-state data using the same linear model as described in the text. This analysis revealed no significant differences between the outcome groups, suggesting that the observed network change is detectable only when collapsing across multiple paradigms rather than during rest.”

Second, I am concerned about the specificity of the identified subnetwork. In the NBS-analysis, the authors use a rather strict $p < 0.0005$ threshold to identify suprathreshold links and still find an extended network of 84 edges among 62 nodes encompassing a large part of the brain. Is there possibly a global connectivity effect driving this result, rather than a specific abnormality of the links in the reported subnetwork? If potential global connectivity differences are mitigated, is there still a significant group-difference in functional connectivity of the NBS-network? Also, if a permutation analysis is performed in which clusters of 84 edges are repeatedly selected at random, how often does this result in significant group-effects?

We thank the reviewer for raising this important question. We would like to demonstrate the specificity of our findings to the identified network in three respects. First, the NBS analysis was performed on the 1st principal component matrices across all paradigms. The PCA step that preceded the NBS analysis rescaled the original connectivity matrices and centered the mean of the derived principal component matrices at zero. As a result,

there should not be a global connectivity effect since the input matrices for all individuals had the exactly same global mean (which equaled zero). Second, to assess whether the *original* connectivity matrices might have different global means, we further examined the group effects on the means of original connectivity matrices for each of the five paradigms. Our result revealed no significant group differences for any of the paradigms ($P > 0.19$), suggesting that the detected network is unlikely to be influenced by global connectivity differences. Third, as suggested, we further performed a permutation test to examine the specificity of the 84 edges identified in this study. During each permutation, we randomly selected 84 edges from the 1st principal component matrices and compared the group differences on the mean of these selected edges. The whole procedure was iterated 10,000 times. We found that none of the P values derived from the 10,000 permutations reached statistical significance after Bonferroni correction (Supplementary Fig S4). In stark contrast, the observed network was highly significant even after Bonferroni correction for the 10,000 permutations. This finding supports the specificity of the identified network in psychosis prediction, demonstrating that it is not driven by effects at the global level. These supplementary analyses are now included in the Supplementary Materials.

[Supplementary Results, Page 7, Specificity of the observed network]: “Since the identified network included a total of 84 edges, the relatively large size of this network raises the question as whether such change was edge-specific or rather generic across the whole brain. Here, we performed an additional permutation test to examine the specificity of the identified network. Specifically, during each permutation, we randomly selected 84 edges from the 1st PC matrices and compared the group differences on the mean of these selected edges. The whole procedure was iterated 10,000 times. We found that none of the P values derived from the 10,000 permutations reached statistical significance after Bonferroni correction (Supplementary Fig S4). In stark contrast, the observed network was highly significant even after Bonferroni correction for the 10,000 permutations. This supplementary analysis supports the specificity of the identified network in psychosis prediction, demonstrating that it is not driven by effects at the global level.”

[Supplementary Fig S4, Page 13]: “Permutation testing on the specificity of the observed cerebello-thalamo-cortical network. For a total of 10,000 permutations (x-axis), none of the derived P values were significant after Bonferroni correction. In stark contrast, the observed network (upper right dot) was highly significant even after Bonferroni correction, suggesting the specificity of the observed network in psychosis prediction. The red dashed line indicates the level of significance.”

Third, the authors report increased functional connectivity between cerebellum, thalamus, and cerebral cortex in CHR subjects and SZ patients. Reviews on functional connectivity in schizophrenia suggest that FC is more commonly found to be reduced, or to involve a combination of increased and decreased FC, in patients relative to controls (e.g., Petterson-Yeo et al., 2011; Fornito et al., 2012), a notable example being a previous publication by the same group reporting increased connectivity between thalamus and sensorimotor cortex, but decreased connectivity among thalamus, prefrontal cortex and cerebellum in the NAPLS 2 sample (Anticevic et al., 2015). The authors do not address the issue of increased versus decreased functional connectivity in schizophrenia/high-risk in general and in relation to cerebello-thalamo-cortical connectivity specifically. Can the authors expand on this issue and comment on possible reasons that they find increased connectivity when decreased connectivity (or a mixture of increased and decreased connectivity) is more often reported? Are there methodological factors that may explain the direction of the effects?

We thank the reviewer for this valuable suggestion. Indeed, the previous work using the resting state data from the same dataset (Anticevic et al., JAMA Psychiatry, 2015) reported a mixed effect (both increased and decreased) for the thalamo-cortical network in subjects at CHR, which seems to be inconsistent with the findings in our current work at first glance. However, we would like to point out two major differences between these two studies, which make the direct comparison of the two studies less straightforward. First, in Anticevic's paper, the analysis was performed on the resting-state data only, and thus the resulting network changes are more likely to reflect *the most significant abnormality in CHR cases during rest*. In contrast, our current paper concerns a different situation – that is, we aimed to investigate *the most consistent functional changes in CHR cases,*

regardless of paradigm. In other words, the observed network change in the current work may not reach statistical significance when examining each paradigm separately (which is actually evidenced in our response to Point #1 above showing that NBS analysis on resting-state data only did not reveal any significant group differences); however, such change can be consistently observed in multiple paradigms. As a consequence, we interpret the results as a reflective of “state-independent” change rather than a significant change during any specific paradigm. Second, from a methodological perspective, the two studies are completely different. Anticevic’s work was based on a hypothesis specifying altered connectivity between thalamus and other parts of the brain, using the thalamus as a seed region. The current work, however, employed a data-driven method (i.e. NBS) without any a priori hypothesis. The NBS method searches all edges across the entire brain and identifies changes that are significant at the whole-brain level. Taken together, Anticevic’s work was a verification study testing a specific hypothesis during resting state, while our current work is a discovery study investigating the most consistent changes in CHR independent of paradigm. We have now included these remarks in the Supplementary Discussion to help clarify the uniqueness of our results.

[Supplementary Discussion, Page 9, Differences between current work and previous work in psychosis CHR]: “The current work differs from previous work (in particular Anticevic et al.) in terms of both methods and interpretation. First, the analysis in Anticevic et al. was performed on the resting-state data only, and thus the resulting network changes are more likely to reflect the most significant abnormality in CHR cases during rest. In contrast, the current work aimed to investigate the most consistent functional changes in CHR cases, regardless of paradigm. As a consequence, we interpret the results as a reflective of “state-independent” change rather than a significant change during any specific paradigm. Second, Anticevic’s work was hypothesis-driven and specifically tested the connectivity between thalamus and other parts of the brain using the thalamus as a seed region. The current work, however, employed a data-driven method (i.e. NBS) without any a priori hypothesis. Taken together, work by Anticevic et al. was a verification study testing a specific hypothesis during resting state, while the current work is an discovery-oriented study investigating the most consistent changes in CHR independent of paradigm.”

Reviewer #2 (Remarks to the Author):

This article deal with an analysis of functional connectivity in a wide dataset of psychiatric disorders, including converter patients. The results are then replicated

in a different dataset, also including psychiatric disorders. Overall the article is clearly written, and the goals and results well presented. Being more familiar with the methodological part than the clinical part, my questions and remarks will mostly focus on the methodological aspect of the manuscript.

We thank the reviewer for your positive feedback on our work! Below please find our response to your questions and concerns.

- First of all, I do not I fully agree with the denomination "cross-paradigm connectivity" used throughout the manuscript. It appear that such a denomination would involve some 'paradigm-specific' connectivity, whereas it appears the way the analysis are done results in exactly the opposite, i.e. get rid of everything linked to the paradigms by regressing out the task regressors. Please justify this denomination, or use another term.

- Apart from the denomination, I am also not sure of the meaning from a methodological point of view of correlating the residuals of a task-based experiment. It would appear to me that all the task-based paradigm where task regressors have been regressed would corresponds to some 'resting-state' effect, i.e. where the focus is based on the baseline rather than the task part. However, it will depend on the design of each specific experiment to assess whether this is really what is measured here: if for example the task consists of a very fast repetition of stimuli over the entire run, then the baseline will be very poorly modeled, and thus the residuals after task regression will consist mostly of noise. Please justify the quality of the baseline evaluation in order to use the method on the chosen paradigms.

We thank the reviewer for raising these very interesting questions. The above two points both relate to the concern as whether the data after regressing out task paradigms would still possess a "task-based" structure or simply reflect a "resting-state" effect. We will respond to both points together here.

First, we would like to clarify that by "regressing out tasks" we only removed the onsets and offsets of each block using boxcar-shaped regressors. The trial-to-trial variability, which is the main source of interregional functional connectivity, was largely preserved in the residual data. As a consequence, the residual time series after task regression was not equal to the "baseline" time series but still possessed trial-to-trial fluctuations that were closely associated with the task structures. The reason to perform such "task regression" analysis has been extensively discussed in the previous studies (Gavrilescu et al., 2008; Horwitz, 2003; Jones et al., 2010; Cao et al., 2014). In particular, compared to resting state, fMRI time series from active tasks are impacted by additional sources of signal variability linked to the temporal structure of the experiment. These

task-dependent signal fluctuations may complicate the interpretation of the resulting functional connectivity estimates, since the systematic “ramping up and down” of BOLD signals in response to the on- and offsets of block conditions may lead to an artificial inflation of the derived connectivity measures. Consequently, the removal of block-to-block signal fluctuations has been proposed as a basic correction step in task-fMRI studies (Gavrilescu et al., 2008; Meyer-Lindenberg, 2009; Muller et al., 2011).

Second, we argue that task regression would not change the fundamental “task-based” architecture of the data, and thus would not drive the residual time series similar to resting state. We demonstrate this using the dataset in this study. Specifically, we tested the similarity between the functional connectivity matrices derived from different paradigms and different processing methods (i.e., rest data vs task data without task regression, rest data vs task data with task regression, task data without task regression vs task data with task regression) by computing their Pearson correlations. If the task regression results in a residualized component resembling the resting state, a close similarity between rest data and task data with task regression would be found, which should be higher than the similarity between task data with task regression and task data without task regression. However, the results showed exactly the opposite - the task data with task regression and task data without task regression were highly correlated ($r > 0.97$), and were much more similar to each other than task data vs rest data ($r < 0.59$, Supplementary Fig S5). These results support the argument that task regression would not drive the residual data to resemble resting state.

Third, to further verify that the task regression step would not change the main hyperconnectivity findings, we reran the entire analysis using the task data without task regression. Our analysis revealed a very similar network covering the cerebellum, thalamus and cerebral cortex ($P_{FWE} = 0.004$, Supplementary Fig S6), suggesting that the observed cerebello-thalamo-cortical hyperconnectivity is not influenced by the task regression procedure. We have now included these supplementary analyses in the Supplementary Materials.

[Supplementary Results, Page 8-9, Validity of using task regression in the study]: “Following the same procedure in the previous work, we regressed out task-related coactivations for the task data. This raises the question as whether the residual data would simply reflect “resting-state” network structure, in which case the claim of “cross-paradigm connectivity” would be invalid. To examine the validity of this processing step, we performed two supplementary analyses. First, we tested the similarity between the

functional connectivity matrices derived from different paradigms and different processing methods (i.e., rest data vs task data without task regression, rest data vs task data with task regression, task data without task regression vs task data with task regression) by computing their Pearson correlations. If the task regression results in a residualized component resembling the resting state, a close similarity between rest data and task data with task regression would be found, which should be higher than the similarity between task data with task regression and task data without task regression. However, the results showed exactly the opposite - the task data with task regression and task data without task regression were highly correlated ($r > 0.97$), and were more similar to each other than task data vs rest data ($r < 0.59$, Supplementary Fig S5). These results support the argument that task regression would not drive the residual data to resemble the resting state.

Second, to assess whether the task regression step would change the main hyperconnectivity findings, we reran the entire analysis using the task data without task regression. Our analysis revealed a very similar network covering the cerebellum, thalamus and cerebral cortex ($P_{FWE} = 0.004$, Supplementary Fig S6), suggesting that the observed cerebello-thalamo-cortical hyperconnectivity is not influenced by the task regression procedure.”

[Supplementary Fig S5, Page 14]: “Similarity between functional connectivity matrices derived from different paradigms and processing methods. The task data with task regression and task data without task regression were highly correlated, and were more similar to each other than the rest data vs task data (regardless of task regression), suggesting that task regression would not drive the residual data to resemble the resting state.”

[Supplementary Fig S6, Page 15]: “Similar cerebello-thalamo-cortical network hyperconnectivity was found using data without task regression.”

● SM ● VIS ● AUD ● DMN ● FPN ● CON ● ATT ● SC-CRB

Gavrilescu et al., Functional connectivity estimation in fMRI data: influence of preprocessing and time course selection. *Human Brain Mapping*, 2008;
 Horwitz B, The elusive concept of brain connectivity. *Neuroimage*, 2003;
 Jones et al., Sources of group differences in functional connectivity: an investigation applied to autism spectrum disorder. *Neuroimage*, 2010;
 Cao et al., Test-retest reliability of fMRI-based graph theoretical properties during working memory, emotion processing, and resting state. *Neuroimage*, 2014.
 Meyer-Lindenberg A, Neural connectivity as an intermediate phenotype: brain networks under genetic control. *Human Brain Mapping*, 2009;
 Muller et al., Underconnected, but how? A survey of functional connectivity fMRI studies in autism spectrum disorders. *Cerebral Cortex*, 2011.

-Finally, it is necessary (as always in functional connectivity) to be extra cautious about the head movement. Here it appears that extra care has been taken in the computation of the connectivity matrices; however it appears that there is a significant difference in the head motion between the groups in the first dataset (table S1, last line). Is the head motion included when testing for differences in overall connectivity between the groups (for example in Figure 2B, or even 2C)? Head motion may indeed lead to higher values in functional connectivity. Please make sure this effect are taken into account in all analyses.

We thank the reviewer for the suggestions on head motion. As head motion is an important confounding factor for connectivity analysis in general, we did make a great effort to control and mitigate its potential effect in the study. These efforts included: 1) all analyses performed in this study incorporated head motion parameters as nuisance regressors (24 head motion parameters and frame-wise displacement); 2) all results discovered in the overall samples were further tested in subsamples in which head motion parameters were matched between groups. These steps ensured that all findings reported in this study were not driven by group differences in head motion. As a further proof, we examined potential associations between head motion (frame-wise displacement) and the observed network metrics across all three groups using Spearman rank-order correlations. This analysis revealed no significant correlation between these two variables ($R = 0.08$, $P = 0.17$), supporting the argument that the detected network abnormality is unlikely to be driven by head motion differences between groups.

[Supplementary Results, Page 8, Association with head motion parameters]:
“To further ensure that the detected network abnormality was not driven by head motion differences between groups, we performed an additional analysis to test the potential association between the observed network metrics and frame-wise displacement values across all individuals in the NAPLS2 sample using Spearman rank-order correlation. This analysis revealed no significant correlation between the two variables ($R = 0.08$, $P = 0.17$), which supports the argument that the detected network abnormality is unlikely to be driven by head motion differences between groups.”

- And a last remark: I am not fully satisfied with the term 'Cerebello-thalamo-cortical': although I understand that I may come from theories in psychiatry, the results shown here do not involve only connection between one of this subpart of the brain: many connections found here between areas within the cortex. Also it make sense to when taking about thalamo-cortical loops (i.e. involving connections between the cortex and the thalamus and the reverse), or even cerebello-thalamo-cortical loops, 'Cerebello-thalamo-cortical' may as well refer to 'whole brain' (apart from some sub-cortical parts of it). Please change the denomination or justify it.

We understand the reviewer’s concern regarding the terminology of “cerebello-thalamo-cortical” loop, and we agree that some of the edges in the identified network were between cortical regions rather than between cortical and subcortical regions. However, out of all 84 observed edges, 25 edges (30%) were connected with cerebellum, and 33 edges (40%) were connected with thalamus, which makes these two interpretable as the hub regions in the identified network. Here, we used the term “cerebello-thalamo-cortical” to emphasize the importance of these two key areas. This term, however, does not preclude potential connections between separate cortical regions. On the contrary, neuroscientists and psychiatrists frequently use this term to refer to a broad pathway system that covers multiple regions in the cerebral cortex (for instance, Shinoda et al., 1993; Andreasen et al., 1998; Palesi et al., 2015; Bernard et al., 2017). In addition, “cerebello-thalamo-cortical circuitry” is also a well-established term for the “cognitive dysmetria” theory of schizophrenia, which relates to the abnormalities in connectivity in the circuitry that links multiple cortical regions, thalamus, as well as cerebellum (Andreasen et al., 1998). As a result, the “cerebello-thalamo-cortical” loop refers to a broad concept rather than exclusive connections between a typical cortical region, thalamus and cerebellum.

Shinoda et al., Thalamocortical organization in the cerebello-thalamo-cortical system. Cerebral Cortex, 1993;

Andreasen et al., "Cognitive dysmetria" as an integrative theory of schizophrenia: a dysfunction in cortical-subcortical-cerebellar circuitry? *Schizophrenia Bulletin*, 1998;

Palesi et al., Contralateral cerebello-thalamo-cortical pathways with prominent involvement of associative areas in humans in vivo. *Brain Structure and Function*, 2015;

Bernard et al., Cerebello-thalamo-cortical networks predict positive symptom progression in individuals at ultra-high risk for psychosis. *Neuroimage: Clinical*, 2017.

Minor notes:

- *The reference 17 (DSM-5) appears very strangely formatted.*

Thank you for pointing this out. The format has been changed according to the format requirement of *Nature Communications*.

- *In the results part, just before the discussion: "The same group effect was again identified ($P = 0.016$, Fig S3B), which was again driven by the differences between the SZ group and the HC group ($P_{Bonferroni} = 0.043$) but not between the other groups ($P_{Bonferroni} > 0.06$)." Do the last stats involve separately different pairs of groups? Why is there a single value in this case?*

The last value ($P_{Bonferroni} > 0.06$) referred to the statistical comparisons of all other group pairs rather than schizophrenia vs controls (i.e., schizophrenia vs bipolar, schizophrenia vs ADHD, bipolar vs ADHD, bipolar vs controls, ADHD vs controls). Since there was a total of five comparisons, we here only reported the minimal P value derived from all these comparisons. That is, P values from all these five comparisons were larger than 0.06.

REVIEWERS' COMMENTS:

Reviewer #1 (Remarks to the Author):

All my concerns have been addressed satisfactorily. I congratulate the authors on a very interesting publication.

Reviewer #2 (Remarks to the Author):

I am fine with the answers provided by the authors